# Roles of Stress Response in Autophagy Processes and Aging-Related Diseases

**DOI:** 10.3390/ijms241813804

**Published:** 2023-09-07

**Authors:** Yoshihisa Watanabe, Katsutoshi Taguchi, Masaki Tanaka

**Affiliations:** 1Department of Basic Geriatrics, Graduate School of Medical Science, Kyoto Prefectural University of Medicine, Kawaramachi-Hirokoji, Kamikyo-ku, Kyoto 602-8566, Japan; 2Department of Anatomy and Neurobiology, Graduate School of Medical Science, Kyoto Prefectural University of Medicine, Kyoto 601-0841, Japan; ktaguchi@koto.kpu-m.ac.jp (K.T.); mtanaka@koto.kpu-m.ac.jp (M.T.)

**Keywords:** macroautophagy, chaperone-mediated autophagy, microautophagy, stress response, proteostasis network, aging

## Abstract

The heat shock factor 1 (HSF1)-mediated stress response pathway and autophagy processes play important roles in the maintenance of proteostasis. Autophagy processes are subdivided into three subtypes: macroautophagy, chaperone-mediated autophagy (CMA), and microautophagy. Recently, molecular chaperones and co-factors were shown to be involved in the selective degradation of substrates by these three autophagy processes. This evidence suggests that autophagy processes are regulated in a coordinated manner by the HSF1-mediated stress response pathway. Recently, various studies have demonstrated that proteostasis pathways including HSF1 and autophagy are implicated in longevity. Furthermore, they serve as therapeutic targets for aging-related diseases such as cancer and neurodegenerative diseases. In the future, these studies will underpin the development of therapies against various diseases.

## 1. Introduction

Stress response and the three autophagy processes individually protect cells against harmful proteins. Recent studies have revealed that both systems function to coordinate protein homeostasis. Molecular chaperones and co-factors also contribute to the selective recognition of substrates in autophagy processes. Impairment of the stress response and autophagy processes is implicated in various diseases such as neurodegenerative diseases and aging, while the acceleration of these processes promotes cancer cell growth. This article reviews the link between the three autophagy processes and the stress response, discussing longevity and current therapeutics against these systems for treating various diseases.

## 2. Autophagy Processes

Autophagy plays important roles in nutrient recycling, the removal of damaged proteins and organelles, and other biological processes. Autophagy pathways are divided into three subtypes: macroautophagy, chaperone-mediated autophagy (CMA), and microautophagy [1]. Macroautophagy involves cell components being sequestered in double-membrane-bound compartments called autophagosomes and then degradation by lysosomes (Figure 1). Formation of the autophagosome is usually mediated by two ubiquitin-like conjugation systems involving Atg12–Atg5 and Atg8 /LC3–phosphatidylethanolamine (PE) conjugate [2]. This sequestration into an autophagosome occurs non-selectively (bulk) or selectively. Usually, cytosolic proteins and organelles such as mitochondria are degraded by bulk macroautophagy upon nutrient starvation in mammals, whereas protein aggregates, damaged organelles and invading bacteria are eliminated by selective macroautophagy [3]. Macroautophagy receptor proteins such as p62/SQSTM1, NBR1, and NDP52 are involved in the selective sequestration of these substrates. Macroautophagy receptors possess the LIR motif consisting of [Trp/Phe/Tyr]–X–X–[Ile/Leu/Val], which binds to the autophagosome marker Atg8/LC3 family of proteins [4]. More than 25 macroautophagy receptors have been identified in mammals. Each macroautophagy receptor binds to its corresponding substrates. For example, protein aggregates bind to macroautophagy receptors p62, NBR1, and OPTN via their ubiquitin binding domain, and are then sequestrated into autophagosomes through interaction with LC3 [5,6,7]. Autophagic clearance of various organelles is facilitated by distinct macroautophagy receptors. Macroautophagy receptors OPTN, NDP52, TAX1BP1, p62, and AMBRA1 are involved in the autophagic elimination of damaged mitochondria called mitophagy [4]. The damaged endoplasmic reticulum is eliminated by FAM134B, SEC62, RTN3, CCPG1, ATL3, and TEX264 [4]. Moreover, damaged lysosomes are targeted by p62 and TRIM16 in selective macroautophagy [8,9]. Invading bacteria are selectively eliminated by macroautophagy receptors p62, NDP52, OPTN, and TAX1BP1 in a similar manner to mitophagy [10,11,12]. 

Apart from nutrient recycling and metabolism of damaged proteins and organelles, macroautophagy has multiple physiological roles including immunity, development, gene regulation, and memory formation. Impairment of macroautophagy causes various diseases such as cancer, neurodegenerative diseases, and immune diseases [13]. Many macroautophagy-related genes have been identified as causative genes of these diseases. For example, monoallelic deletion of the gene encoding Beclin1, a central regulator of macroautophagy, is detected in 40–75% of sporadic human breast and ovarian cancers [14]. Mutations of Parkin and PINK1 cause early-onset familial Parkinson’s disease [15,16]. Both proteins are involved in selective macroautophagic clearance of damaged mitochondria [17]. Moreover, the genes encoding macroautophagy receptors p62 and OPTN are reported as causative genes of amyotrophic lateral sclerosis and Paget’s disease of bone [18,19,20,21]. Additionally, macroautophagy is associated with metabolic syndrome and aging, and it is an important therapeutic target for various diseases.

Unlike macroautophagy, CMA mediates the direct incorporation of substrates into lysosomes (Figure 1). This incorporation takes place via molecular chaperone HSC70 and its co-factors, such as CHIP, HOP, and HSP40 [22]. The HSC70 complex binds to specific target sequences of substrate proteins, Lys–Phe–Glu–Arg–Gln (KFERQ-like motif), and substrates are then transported to the lysosomal receptor LAMP2A. Substrates are unfolded by HSC70 and translocated into the lysosome with the help of lysosomal HSC70 [23]. Ribonuclease A, glyceraldehyde-3-phosphate dehydrogenase and α-synuclein are selective substrates of CMA [24,25]. A recent study revealed that about 45–47% of human proteins are potential substrates for CMA [26]. Although CMA constitutively occurs under physiological conditions, oxidative and hypoxic stresses induce its activation in a similar manner to macroautophagy [27,28]. However, in contrast to macroautophagy, CMA cannot degrade aggregated proteins [25,29]. Thus, CMA functions as an initial step in the defense against cytotoxic proteins that are aggregate-prone.

CMA contributes to other physiological roles. As observed for macroautophagy, CMA is involved in nutrient recycling under starvation conditions. However, CMA is induced by nutrient starvation later than macroautophagy [22]. Moreover, CMA regulates metabolic pathways including glucose and lipids through the degradation of metabolic enzymes. A liver-specific conditional knockout mouse of LAMP2A has phenotypes of reduced hepatic glycogen storage and hepatic steatosis [30]. A proteomics analysis revealed that 30% of CMA substrates are lipid metabolic enzymes of triglyceride synthesis, steroid metabolism, and lipid binding and transport [30]. CMA also regulates the immune system. The ubiquitin ligase Itch and the calcineurin inhibitor RCAN1, which are negative regulators of T cell receptors, are selectively degraded by CMA in activated T cells [31]. LAMP2A-deficient CD4^+^ T cells show significantly less activation-induced cell proliferation and lower cytokine expression [31]. Apart from this, STING, a central signaling molecule in the innate immune response to cytosolic nucleic acids, is desumoylated by Senp2 and subsequently degraded by CMA in the late phase of viral infection, resulting in deactivation of innate immune responses [32].

Impairment of CMA leads to various diseases. Parkinson’s disease-related proteins such as α-synuclein, DJ-1, and LRRK2 are selective substrates for CMA [25,33,34]. The pathogenic mutants of α-synuclein interfere with LAMP2A and reduce the degradation of CMA substrates [25]. In addition, CMA is also reported to degrade TDP-43 and huntingtin (Htt), which are linked to amyotrophic lateral sclerosis (ALS) and Huntington’s disease (HD), respectively [35,36]. These observations suggest that impairment of CMA might be the cause of various neurodegenerative diseases. CMA is also associated with cancer. In various cancer cells, CMA is activated through the up-regulated expression of LAMP2A and is required for malignant cell growth and tumor metastasis [37]. CMA degrades some glycolytic enzymes, anti-proliferation proteins, tumor suppressor proteins, and pro-apoptotic proteins, resulting in the promotion of growth and proliferation [38,39,40]. Therefore, CMA is expected to be a potential target for cancer therapy.

Lysosomes can directly engulf soluble materials by invagination and pinching off of lysosomal membranes for their degradation (Figure 1). This catabolic process is called microautophagy. Microautophagy plays roles in nutrient recycling, lipid metabolism, maintenance of organellar size, and membrane homeostasis [41]. Recent studies proposed that microautophagy is categorized into three distinct types [42]. The lysosomal membrane wraps substrates (protrusion). Atg18 is required for this process in yeast [43]. The second type of microautophagy is direct engulfment by invagination of the lysosomal membrane (invagination). In yeast, this pathway functions by degrading cytoplasmic components, ER, mitochondria, portions of the nucleus, and lipid droplets [42]. Multiple Atgs are also required for invagination of the lysosomal membrane [44]. The third type is a microautophagic degradation pathway through late endosomes/multivesicular bodies (MVBs). Cytoplasmic cargo proteins selectively bind to HSC70 and macroautophagy receptor Nbr1 and are incorporated into late endosomes/MVBs. This process takes place in endosomal sorting complexes required for transport (ESCRTs)-dependent manner [45,46]. MVBs are fused to lysosomes/vacuoles, and cargos are degraded in lysosomes. Three distinct autophagic pathways play roles in numerous physiological processes and are regulated coordinately by various stress proteins such as molecular chaperones for maintaining cellular homeostasis.

## 3. Role of Stress Response Pathway in Macroautophagy

Many physiological stresses generate abnormal proteins that are the cause of various diseases. To protect against abnormal proteins, heat shock proteins (HSPs) including molecular chaperones play important roles in maintaining cellular homeostasis. Heat shock factor 1 (HSF1), a master regulator of heat shock response, controls the expression of HSPs, ubiquitin proteasome system-related genes, and macroautophagy-related genes. Heat shock stress induces macroautophagy in neuroblastoma, hepatoma, and cardiomyocytes [47,48,49]. Heat stress at 42℃ caused a rapid increase in protein levels of Beclin 1, Atg7, and Atg5-Atg12 complex in HeLa cells [50]. Simultaneously, conversion of LC3-I (inactive form) to LC3-II (active form) was induced after 5 min of heat shock, suggesting that macroautophagy is closely linked to the heat shock response network [50].

Various heat shock proteins are involved in the activation of macroautophagy. HSPB8, a small heat shock protein, stimulates the degradation of aggregation-prone proteins, such as polyglutamine protein Htt43Q and superoxide dismutase 1, by macroautophagy, in coordination with the co-factor BAG3 [51,52]. Although BAG3 is known as a nucleotide exchange factor of HSP70, the interaction between BAG3 and HSP70 is not required for macroautophagy activity [53]. Thus, the HSPB8-BAG3 complex recognizes harmful proteins, and these proteins are directed to macroautophagic clearance. Similarly, the interaction of p62 with BAG3 also regulates macroautophagy flux [54]. p62 is composed of a Phox/Bem domain 1 (PB1), a ZZ-type zinc finger domain, an LC3-interacting region (LIR) motif, a Keap1-interacting region (KIR), and a ubiquitin-associated domain (UBA). Ubiquitinated proteins are trapped by p62 via its UBA domain and oligomerize through the PB1 domain sequentially. Phosphorylation of p62 is required for this process [55,56,57]. Interestingly, phosphorylation of p62 is reduced in HSF1 knockout cells and HSF1 inhibitor-treated cells, leading to a delay in the elimination of harmful proteins [57]. Furthermore, p62 phosphorylation is also reduced by knockdown of HSPB8 and BAG3 [58]. These observations indicate that HSF1-mediated stress response is required for p62 activation.

Recently, chaperonin containing TCP1 subunit 2 (CCT2) was reported as a novel macroautophagy receptor for solid protein aggregates [59]. CCT is a member of the chaperonin family of molecular chaperones and is composed of eight different subunits forming a double-stacked ring structure [60]. CCT is known to participate in the folding of newly synthesized tubulin and actin proteins [61], while it also plays a role in the macroautophagic clearance of mutant ATXN3 or Htt aggregates [62]. CCT2 is an essential component of the molecular chaperone CCT. However, CCT2’s function is switched from a chaperonin to a specific aggrephagy receptor when excessive protein aggregates are accumulated [59]. CCT2 solely binds to aggregates in a ubiquitin-independent manner and mediates their sequestration into autophagosomes [59]. 

Apart from HSPs, HSF1 is involved in the expression of various genes encoding macroautophagy-related factors. For example, HSF1 directly binds to *Atg7* and *Atg4B* promoters and upregulates their expression [63,64]. Both genes play a critical role in the macroautophagy pathway. Atg7 is a macroautophagy-related E1 ligase and activates LC3 and Atg12. Atg4B cleaves the precursor product of LC3 in the C-terminal region, thereby initiating autophagosome formation [65]. Other than the core machinery, the expression of various macroautophagy-associated proteins is also driven by HSF1. ChIP-seq analysis revealed that both HSF1 and HSF2 bind the *CCT2* gene under heat shock conditions [66]. Moreover, BAG3, p62, and HSPB8 expression are also regulated by HSF1 [67,68]. These observations show that the HSF1-mediated stress response pathway induces both protein refolding and activation of p62 via molecular chaperones and co-factors including HSPB8, HSP90, and BAG3 (Figure 2). Subsequently, irreversible protein aggregates bind to the activated p62-HSPB8-BAG3 complex and are engulfed into autophagosomes (Figure 2). Collectively, HSF1-mediated stress response contributes to maintaining proteostasis through protein refolding and macroautophagy. 

## 4. Role of Molecular Chaperones in CMA and Microautophagy

CMA and microautophagy have a selective degradation process that is similar to macroautophagy. HSC70 plays an important role in both pathways. As described above, HSC70 recognizes a specific target sequence of substrate proteins, KFERQ-like motif, in the CMA pathway and binds to LAMP2A at the lysosomal membrane (Figure 1). Similarly, HSC70 selectively binds to the KFERQ-like motif of substrates in the endosomal microautophagy pathway. HSC70-cargo complexes directly bind to phosphatidylserine at the endosomal membrane via the HSC70 LID domain, and are then sequestered into endosomes [69,70]. The HSC70-mediated microautophagy pathway is reported to regulate the efficiency of neurotransmitter release in presynaptic terminals through local protein quality control [71]. This evidence shows that this pathway might impact on various brain functions such as memory and neural circuit formation.

In addition to HSC70, other molecular chaperones also participate in the degradation of individual substrates by CMA and microautophagy. Cell death is a critical process for the development and homeostasis of almost all multicellular organisms. Cell death has been divided into apoptosis and non-apoptotic cell death. Ferroptosis was proposed as an iron-dependent and non-apoptotic cell death by Dixon et al. [72]. Extracellular Fe^3+^ is bound by transferrin, and endocytically internalized by the transferrin receptor [73]. Within the cytosol, Fe^3+^ is reduced to Fe^2+^, and Fe^2+^ is temporally stored in the labile iron pool and ferritin [73]. An increase in free intracellular Fe^2+^ generates highly reactive hydroxyl radicals through the Fenton reaction, inducing lipid peroxidation in the cell membrane [74]. Excessive lipid peroxidation leads to cell death by ferroptosis. The antioxidant enzyme glutathione peroxidase 4 (GPX4) reduces lipid peroxidation and inhibits ferroptosis [75]. Zheming et al. recently described that CMA mediates the degradation of GPX4 in an HSP90-dependent manner and induces ferroptosis [76]. Although the role of HSP90 in CMA is not well defined, HSP90 participates in the ferroptosis process by regulating the levels of LAMP2A in CMA [76]. Interestingly, the HSP90 inhibitor CDDO inhibited erastin-induced ferroptosis in a hippocampal-derived HT-22 cell line [76]. Ferroptosis is closely related to many diseases, such as cancer, nervous system diseases, ischemia-reperfusion injury, kidney injury, and blood diseases. Thus, both CMA and HSP90 may be potential therapeutic targets for various diseases.

DnaJ heat shock protein family (Hsp40) member C5 (DNAJC5) promotes ESCRT-dependent microautophagy. Misfolded cytosolic proteins are selectively exported to the extracellular milieu via an unconventional secretory path termed Misfolding-Associated Protein Secretion (MAPS) and translocated into MVBs. DNAJC5 chaperones misfolded proteins to the lumen of endolysosomes via ESCRT-dependent microautophagy [77]. Mutations in *DNAJC5* are implicated in adult neuronal ceroid lipofuscinosis (ANCL), a dominantly-inherited neurodegenerative disease [78]. ANCL is characterized by the hallmark lipofuscin deposit (a mixture of lipids and proteins with metal materials) inside the lysosomal lumen, which typically emits auto-fluorescence [79]. Notably, ANCL mutations of *DNAJC5* inhibited MAPS without affecting DNAJC5 endolysosomal translocation, causing lipofuscinosis and neurodegeneration [77]. This result implies that ANCL is caused by an imbalance of two DNAJC5-mediated proteostasis, including MAPS and microautophagy.

CMA and microautophagy are also activated in response to various types of stress. CMA adaptor proteins HSC70 is translocated into the lysosomes after exposure to heat shock (34°C for 1 h) in a cell line derived from the tailfin of the marine teleost yellowtail fish [80], implying the active uptake of degradation substrates into the lysosomes. Moreover, nutrient starvation is reduced the rate of degradation of LAMP2A and increases its reinsertion into the lysosomal membrane of the rat liver [27]. Meanwhile, oxidizing conditions elevate LAMP2A by increasing the rate of transcription of its gene [27]. These indicate that the activation mechanism of CMA is likely to vary depending on the stress. Microautophagy is also induced by oxidative stress and DNA damage in drosophila. When drosophila larvae were treated with the oxidative stressor paraquat or the topoisomerase II inhibitor etoposide, numerous microautophagic puncta were intracellularly detected using a microautophagy sensor [81]. Intriguingly, reducing macroautophagy leads to a compensatory enhancement of microautophagy, suggesting a tight interplay between these degradative processes [81]. Similar to macroautophagy, CMA and microautophagy also contribute to cell protection under stress conditions cooperatively with the HSF1-mediated stress response pathway. Age-related reductions in proteostasis activities, including protein refolding and the degradation of aggregated proteins, causes various diseases.

## 5. Proteostasis and Longevity

The capacity of the proteostasis network, including autophagy processes declines with age, causing the accumulation of protein aggregates and diminished physiological functions [82]. Conversely, enhanced proteostasis provides anti-aging benefits. As shown in Table 1, numerous studies have revealed that proteostasis functions are also linked to longevity [83]. For example, heat-induced expression of HSP70 caused a period of decreased mortality rate in drosophila [84]. Besides this, the expression of small heat shock proteins such as HSP22 and HSP23 correlates with lifespan extension in drosophila lines genetically selected for increased longevity [85]. This evidence suggests that the induction of HSPs protects cells from damaged proteins during aging and contributes to longevity. Similarly, HSF1, a master regulator of HSP genes, has an impact on lifespan extension. In *Caenorhabditis elegans*, reduced expression of HSF1 using RNA interference leads to a shortened lifespan, and its increased expression effects enhanced resistance to heat and oxidative stress, resulting in lifespan extension [86]. HSF1 predominantly exists as an inactive monomer in the cytoplasm under physiological conditions. Upon stress stimuli, HSF1 is activated by a multistep process involving nuclear localization, trimerization, acquisition of DNA-binding, and transcriptional activities, which coincide with several posttranslational modifications [87]. HSF1 undergoes phosphorylation, acetylation, and sumoylation on several residues. Phosphorylation at Ser-230, Ser-326, and Ser-419 is the activating modification, whereas acetylation at Lys-80 is the repressive modification [88]. Sirtuin-1 (SIRT1), a nicotinamide adenine dinucleotide dependent protein deacetylase, is involved in the deacetylation of HSF1 at Lys-80 and enhances HSF1 binding to the heat shock elements that are present upstream of heat shock genes [89]. The sirtuin pathway is well known to regulate lifespan in a wide range of organisms including yeast, nematodes, drosophila, and rodents [90]. SIRT1 is the most studied among the mammalian SIRTs (SIRT 1-7) and regulates physiological processes to enhance lifespan including energy metabolism, mitochondrial biogenesis, cell growth and differentiation, inflammation, and cognitive and memory functions [91,92]. Dietary calorie restriction (CR) in mice induces SIRT1 expression in many tissues such as the brain, visceral fat pads, kidney, and liver of 12-month-old rats fed a CR diet [93]. Enhanced SIRT1-expression promotes the deacetylation of the DNA repair factor Ku70 in various tissues, causing sequestration of the proapoptotic factor Bax away from mitochondria [93]. The link between the sirtuin pathway and aging was originally discovered through analysis of the SIR4-42 longevity mutation of budding yeast [94,95]. SIR2, the yeast SIRT1 ortholog, interacts with SIR4, and its overexpression extends the lifespan of yeast by 50% [96,97]. In rodents, brain-specific SIRT1-overexpression contributed to lifespan extension (∼11% extension) [98]. SIRT1 deacetylates a large number of proteins including histone proteins, transcription factors, and DNA repair proteins [99]. Through SIRT1-mediated deacetylation, SIRT1 associates with the major longevity pathways, for example, AMPK and insulin–IGF1 signaling, including targets such as protein kinase A, mTOR, forkhead box O, and IGF1 [99]. These signaling pathways are closely linked to the nutrient response. Shortage of nutrients including carbohydrates, amino acids, and lipids by CR impacts the longevity pathways. The relationship between CR and lifespan extension has been studied in various species over the years [100]. The effect of CR was recently reverified in rhesus monkeys using two independent studies [101]. This study indicated that CR is effective in delaying the effects of aging, and it prevents the onset of age-related diseases including cancer, diabetes, and cardiovascular disease in primates [101].

CR and fasting affect various cellular processes. Originally, macroautophagy was discovered as a critical function in nutrient homeostasis during amino acid starvation [102]. Under nutrient starvation, macroautophagy is activated to supply nutrients by self-digestion. mTORC1 and AMPK act as nutrient sensors of amino acids and glucose and control macroautophagy activity. In nutrient-rich conditions, mTORC1 is activated and drives anabolic processes including protein, lipid, and nucleotide synthesis, while simultaneously inhibiting catabolic pathways such as macroautophagy [103]. Indeed, the mTORC1 inhibitor rapamycin can induce macroautophagy [104]. AMPK is activated in response to an increase in AMP and ADP levels when the energy state of the cell is compromised [105]. mTORC1 inhibits ULK1 kinase complexes through direct phosphorylation of ULK1 and suppresses autophagosome formation in nutrient-rich conditions [106]. Upon starvation, mTORC1 inactivation allows ULK1 kinase complexes to initiate the formation of autophagosomes [106]. Similarly, mTORC1 also regulates lysosomal function in a nutrient-dependent manner. mTORC1 is localized on the lysosomal surface under nutrient-rich conditions, and then the master regulator of lysosome biogenesis, TFEB, is inhibited by mTORC1-mediated phosphorylation [107]. When cells are exposed to nutrient starvation, the inactivation of mTORC1 leads to rapid translocation of TFEB to the nucleus and activation of lysosomal ATP-sensitive two-pore Na^+^ channels, and subsequently promotes lysosomal biogenesis and activity [108]. Conversely, AMPK activation induces macroautophagy through mTORC1 inhibition and the activation of ULK1 kinase complexes upon starvation [105]. 

Multiple groups have studied the effect of mTORC1 inhibition on the longevity of mice. Oral or intraperitoneal administration of rapamycin is effective in extending lifespan with females typically benefiting more than males [109]. Furthermore, it is interesting to note that more beneficial effects of rapamycin on lifespan extension were observed in various disease model mice (autosomal dominant polycystic kidney disease, multiple different cancers, epilepsy, immunodeficiency, and mitochondrial disease) [109]. Similarly, AMPK activation also improves healthspan and lifespan in mice. The type 2 diabetes medication metformin can activate AMPK by inhibiting complex I of the mitochondrial electron transport chain [110]. Long-term diet supplementation with 0.1% metformin led to a 5.8% extension of mean lifespan in male mice although a higher concentration of the drug (1%) was toxic and significantly shortened mean lifespan [111]. Moreover, low-dose metformin improved the physical health of the animals as an index of locomotor activity such as rotarod, treadmill, and open-field tests [111]. Rapamycin and metformin are well known to activate macroautophagy via mTORC1 and AMPK pathways. These findings raise the possibility that macroautophagy is an integral mediator of lifespan extension [112].

Furthermore, several macroautophagy-related proteins including Atg5, Atg7, and LC3 are acetylated by the p300 acetyltransferase under nutrient-rich conditions [113]. Lee et al. revealed that acetylation of LC3 inhibits macroautophagy and that SIRT1 deacetylates it during starvation [114]. Indeed, overexpression of SIRT1 induced the conversion of LC3-I to LC3-II in HCT116 cells [114]. These findings indicate that both the HSF1-mediated proteostasis, and macroautophagy might be controlled by SIRT1, contributing to the suppression of aging.

Recent studies have demonstrated that constitutive activation of macroautophagy promotes longevity in nematodes, drosophila, and rodents. Overexpression of Atg1, Atg5, and Atg8 in drosophila or mice extends lifespan [115,116,117]. Similarly, mutations of macroautophagy suppressors contribute to longevity. Rubicon and BCL2 interact with Beclin 1, a master regulator of macroautophagy, and suppress its activity. Conversely, genetic disruption of these interactions leads to constitutive activation of basal macroautophagy [118]. Targeted mutant mice with a Phe121Ala mutation in Beclin 1 (*Becn 1*^F121A/F121A^) that decreases its interaction with BCL2 were constitutively increased in basal macroautophagy of multiple organs and showed a significant lifespan extension [119]. Likewise, Rubicon knockdown nematodes and flies showed extended lifespan, and its knockdown in the neurons was most effective [120]. Moreover, treatment with the macroautophagy-inducer spermidine suppressed necrosis and extended the lifespan of yeast, flies and worms, and human immune cells [121]. These findings suggest a positive correlation between macroautophagy and longevity. Interestingly, new findings show that synchronization of macroautophagy with circadian rhythm is essential for lifespan extension. In drosophila, night-specific induction of macroautophagy was both necessary and sufficient to extend lifespan on an ad libitum diet, whereas day-specific induction of macroautophagy did not extend lifespan [122]. Accordingly, circadian-regulated macroautophagy might delay the onset of aging.

**Table 1 ijms-24-13804-t001:** Proteostasis and longevity.

Targets	Functions	Effects on Lifespan Extension	**Ref.**
** *C. elegans* **			
HSF1	Transcription factor	*hsf1 RNAi:* lifespan extension (−37%) *hsf1* overexpression: lifespan extension (+40%)	[86]
EP300	Macroautophagy	Acetyltransferase EP300 inhibitor (macroautophagy activation) lifespan extension (+15%)	[121]
Rubicon	Macroautophagy	*rub-1* knockdown (macroautophagy activation) lifespan extension (+21%)	[120]
** *Drosophila* **			
HSP70	Chaperone	Heat treatment (20–30 min): lifespan extension (+30%)	[84]
HSP22	Chaperone	Long-lived drosophila lines: *hsp22* RNA level (2–10 fold increase)	[85]
Atg1	Macroautophagy	Neuronal overexpression lifespan extension (+25%)	[117]
Atg8	Macroautophagy	Neuronal overexpression lifespan extension (+56%)	[115]
AMPK	Macroautophagy	Neuronal overexpression lifespan extension (+113%)	[117]
EP300	Macroautophagy	Acetyltransferase EP300 inhibitor (macroautophagy activation) lifespan extension (+30%)	[121]
Rubicon	Macroautophagy	*dRubicon* knockdown (macroautophagy activation) lifespan extension (+20%)	[120]
** *Mice* **			
Atg5	Macroautophagy	Overexpression lifespan extension (+17.2%)	[116]
mTORC1	Kinase	mTORC1 inhibitor rapamycin (macroautophagy activation) Oral administrationheterogeneous mice: lifespan extension (males: +9%; females: +14%) C57BL/6J.Nia: lifespan extension (males: +11%; females: +16%) Intraperitoneal administrationC57BL/6J.Nia: lifespan extension (+14%) CD1: (males: +8.9%; females: +8.4%)	[109]
AMPK	Kinase	AMPK activator metformin (macroautophagy activation) Oral administration 0.1% in diet lifespan extension (C57BL/6 males: +5.8%; B6C3F1 males: +4.2%) 1% in diet: lifespan extension (C57BL/6 males: −14.4)	[111]
Beclin 1	Macroautophagy	*Becn1^F121A/F121A^* knock-in mice (macroautophagy activation) lifespan extension (males: +12%; females: +11%)	[119]

SIRT3 is a member of the sirtuin family, and its active form is present in the mitochondrial matrix, where it deacetylates many proteins to regulate diverse mitochondrial functions such as ATP production, reactive oxygen species management, and *β*-oxidation [112]. SIRT3 expression is increased in the liver, skeletal muscle, and adipose tissue during CR [123]. Recent research disclosed that SIRT3 regulates lipid metabolism through macroautophagy and CMA [124]. Starvation and an acute lipid stimulus induce the degradation of lipid droplets. In this process, SIRT3 activates AMPK and increases the macroautophagic sequestration of lipid droplets [124]. Coincidentally, SIRT3 also induces the binding of lipid droplet coating proteins to the HSC70-LAMP2A complex, thereby increasing lipid droplet breakdown [124]. This suggests a potential role of CMA in longevity. Overexpression of PTEN, a negative regulator of the INS/PI3K/AKT pathway, enhances CMA, dependent upon PTEN’s lipid phosphatase activity and AKT inactivation, and leads to lifespan extension [125,126]. 

Similarly to macroautophagy, the activity of CMA declines with aging. There are differences in the binding and uptake of CMA substrate proteins between lysosomes in young (3-month-old) and old (22-month-old) livers [127]. These differences are responsible for the age-related decrease in the levels of LAMP2A, while the cytosolic levels and activity of HSC70 remain unchanged with age [127]. As mentioned above, CMA and microautophagy are also activated upon nutrient starvation. Hence, these processes have the potential to extend lifespan, although there is limited experimental evidence implicating them in longevity. In the future, novel studies might help to elucidate thiis.

## 6. Stress Response and Autophagy Processes as Therapeutic Targets of Neurodegenerative Diseases and Cancer 

Stress response and autophagy processes are controlled in a coordinated manner to maintain cellular homeostasis. The disruption of proteostasis causes various diseases. For example, physiological analysis of systemic *Atg7* or *Atg5* knockout in adult mice showed neurodegeneration, liver abnormalities, splenic enlargement, testicular degeneration, low white adipose tissue, degenerative changes in muscle, small myofibers, and pancreatic damage [128]. Liver-specific *Atg5* or *Atg7* knockout mice develop benign liver adenomas, and ubiquitinated proteins accumulate in the liver [129]. Accumulation of ubiquitinated proteins, apoptosis, and neurodegeneration are observed in neuron-specific knockout mice of these genes [128]. Similarly to these mice, neuron-specific 26S proteasome-depleted mice also show α-synuclein, TDP-43, and FUS pathologies [130,131]. Furthermore, mutations of macroautophagy-related genes are implicated in various genetic diseases [132]. In addition to the above-mentioned Beclin1, Parkin, PINK1, p62, and OPTN, mutations in genes encoding ATG16L1, ATG16L2, ATG5, Clec16a, and CALCOCO2 link to autoimmune diseases and inflammatory diseases such as Crohn’s disease, systemic lupus erythematosus, systemic sclerosis, diabetes, and multiple sclerosis [132]. As for neurodegenerative and neurodevelopmental diseases, WDR45, FAM134B, VPS13D, VPS15, and EPG5 are implicated in neurodegeneration with brain iron accumulation, Vici syndrome, hereditary sensory, and autonomic neuropathy type II and ataxia with spasticity [132]. Interestingly, impaired crosstalk between macroautophagy and the endoplasmic reticulum (ER) stress response contributes to the pathogenesis of Crohn’s disease. The extracellular release of the antibacterial protein lysozyme via the conventional secretion pathway is inhibited in intestinal Paneth cells infected with the intestinal pathogen *Salmonella enterica serovar Typhimurium*, while lysozyme is secreted via a macroautophagy-based alternative secretion pathway called secretory macroautophagy [133]. Secretory macroautophagy was triggered by ER stress and required interleukin-22 from type 3 innate lymphoid cells [133]. In mice expressing a Crohn’s disease-associated risk polymorphism (*ATG16L1^T300A^*), secreted lysozyme is significantly decreased, suggesting that the dysfunction of the ER stress response and ATG16L1-dependent secretory macroautophagy etiologically are associated with Crohn’s disease [133]. 

Danon disease is an X-linked dominant skeletal and cardiac muscle disorder that is caused by loss-of-function mutations in the *LAMP2* gene [134]. *LAMP2*-deficient mice show extensive accumulation of macroautophagic vacuoles in many tissues including the liver, pancreas, spleen, kidney, and skeletal and heart muscles [135]. Alternative splicing of the LAMP2 pre-mRNA leads to three different isoforms, LAMP2A, LAMP2B, and LAMP2C [136]. LAMP2A is a crucial player in CMA as described above. LAMP2B mediates autophagosome-lysosome fusion in cardiomyocytes and acts as a component of exosome membranes [137]. LAMP2C has recently been identified as being involved in the selective autophagic degradation of DNA and RNA molecules [137]. Although most Danon patients carry mutations that result in a deficiency of all three LAMP2 isoforms, LAMP2B dysfunction is thought to be affected in Danon disease [137].

Current research efforts examining these processes are not only focused on the elucidation of their physiological mechanisms but also on the development of therapeutic targets against neurodegenerative diseases and cancer. Pathogenic aggregated proteins, such as α-synuclein, polyglutamine proteins, and TDP43, are degraded by macroautophagy. The activation of the stress response and autophagy processes can contribute to the clearance of these proteins [138,139,140]. For example, the tubulin destabilizing agent colchicine enhances the expression of HSPB8 and several macroautophagy players through the master regulator of macroautophagy TFEB [141]. At present, clinical trials of colchicine for ALS have started in Italy [142]. Tamoxifen is also a macroautophagy inducer, and a clinical trial involving this compound is being conducted for ALS [143]. Furthermore, a novel therapy for HD using the CMA process was developed. This strategy involves specific recognition of mutant Htt proteins by a fusion protein of polyglutamine binding peptide 1 and the KFERQ-like motif and elimination through CMA-mediated degradation [36]. A decrease in polyglutamine aggregation and an increase in life span were observed when this fusion protein was expressed in the brain of an HD mouse model [36]. This novel therapy might also be effective in treating other neurodegenerative diseases.

In contrast to neurodegenerative diseases, the strategy of cancer treatments may be the opposite regarding the targets of autophagy and stress response. Increased levels of HSPs such as HSP70, HSP90, and HSP27 are observed in a number of cancers, and are involved in tumorigenicity, metastatic potential, and the resistance to chemotherapy of cancer cells [144]. Actually, a HSP90 inhibitor, geldanamycin, was reported to have in vitro and in vivo antineoplastic activities using preclinical models [145]. As the clinical trials registered on ClinicalTrials.gov, STA-9090 and XL888, less toxic HSP90 inhibitors, are being evaluated in clinical trials against various cancers, such as ovarian cancer and advanced gastrointestinal cancer [146]. Similarly, HSF1 is also a potential therapeutic target for cancer. HSF1 deficiency suppresses oncogenesis induced by the mutagen dimethylbenzanthracene and p53 or RAS mutations, suggesting that HSF1 assists malignant transformation by the augmentation of proliferation, survival, protein synthesis and glucose metabolism [147]. Currently, clinical trials of the HSF1 inhibitors quercetin and triptolide are being conducted for squamous cell carcinoma, gastric cancer, and breast cancer, although it remains unknown whether these drugs are specific inhibitors of HSF1 [146]. Furthermore, macroautophagy has also received a significant amount of attention as an effective therapeutic target for various cancers. Chloroquine and hydroxychloroquine are being used in clinical trials as therapies against lung cancer, breast cancer, and prostate cancer [146]. 

## 7. Conclusions

The stress response pathway and autophagy processes combine to coordinate the maintenance of proteostasis. Molecular chaperones and co-factors are required not only for protein refolding but also for substrate recognition directed to autophagy processes. However, it remains unknown how autophagy processes are activated by molecular chaperones and whether other molecular chaperones are involved in these processes. A novel proteostasis network will be revealed through these studies. Long-term administration of several macroautophagy activators improves not only lifespan extension but also healthspan. In addition to drugs, a healthy lifestyle, including exercise, fasting and sleep, can enhance systemic macroautophagy, promoting health, and longevity. These aspects are also highly important from a preventive medicine perspective for various diseases. Moreover, clinical trials of HSP90 inhibitors, macroautophagy inhibitors, and macroautophagy inducers that target various cancers and ALS are being conducted by a number of research groups. New findings from clinical trials will provide important information about the efficacy and safety of therapies targeting the proteostasis network. This review focuses on the collaboration between molecular chaperones and autophagy processes, whereas each of them has been evaluated independently for its effectiveness in clinical trials of cancers and neurodegenerative diseases. Clinical benefits might be enhanced by concurrently targeting the functions of both these entities in disease treatment. For example, combining hydroxychloroquine with HSP90 inhibitors may be more effective than using them individually in cancer therapies. The reason is that when each drug inhibits one of the proteostasis processes, the possibility exists that the therapeutic effectiveness might weaken because the other process could become enhanced. The concept extends to neurodegenerative diseases too. Enhancing cellular protection can likely be achieved by simultaneously inhibiting protein aggregation through molecular chaperones and promoting degradation through autophagy. Therefore, it is crucial to comprehensively understand proteostasis functions and translate this knowledge into the development of new therapeutic approaches. In the future, proteostasis research including autophagy processes is expected to play a significant role in the treatment of various diseases associated with aging.

## Figures and Tables

**Figure 1 ijms-24-13804-f001:**
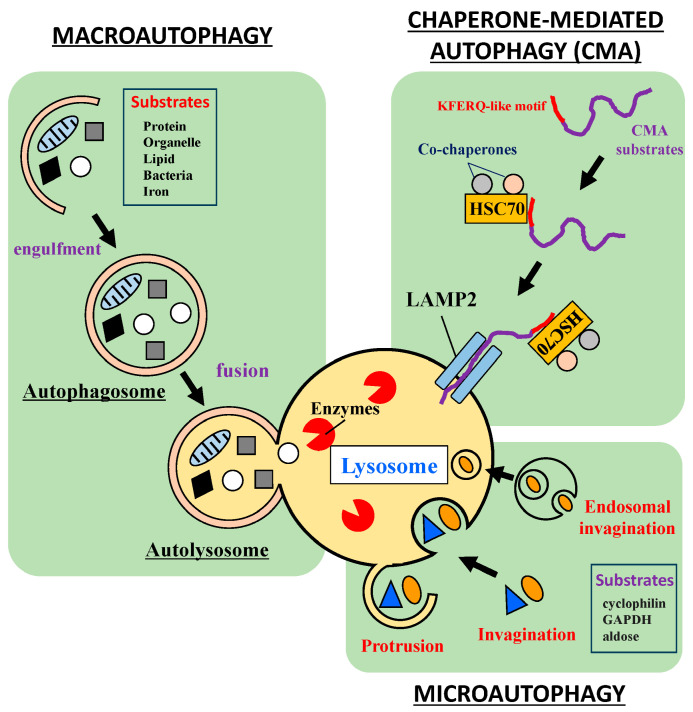
Autophagy processes. The three subtypes of autophagy involved in cellular metabolism and homeostasis include macroautophagy, chaperone-mediated autophagy, and microautophagy. These degradation systems eliminate various substrates including cellular components and invading bacteria.

**Figure 2 ijms-24-13804-f002:**
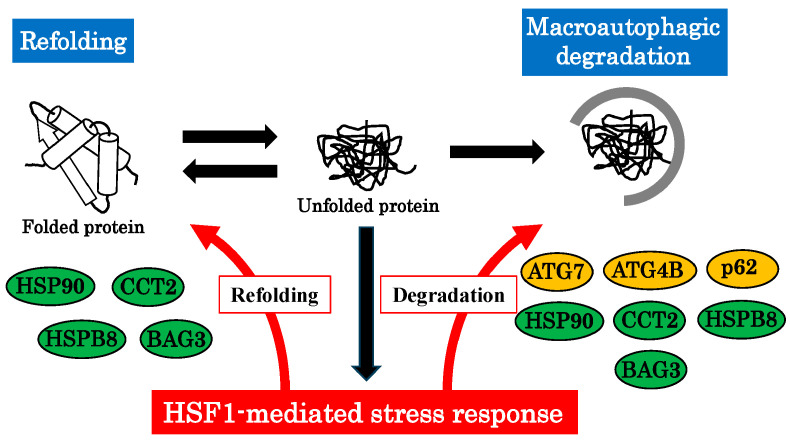
Coordinated proteostasis of the stress response and macroautophagy. Accumulation of harmful proteins by physiological stresses activates the HSF1-mediated stress response pathway, causing an increase in molecular chaperones, co-chaperones, and macroautophagy-related genes. Harmful proteins are refolded by molecular chaperones, which simultaneously activate the macroautophagy adaptor p62. Protein aggregates bind to active p62 and are engulfed into autophagosomes.

## Data Availability

Not applicable.

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
