# Peer review of "Roles of Stress Response in Autophagy Processes and Aging-Related Diseases"

_ijms, 2023, doi:10.3390/ijms241813804_

Round 1

Reviewer 1 Report

I reviewed the submitted review article by Watanabe et al., titled "Roles of Stress Response in Autophagy Processes and Aging-Related Diseases" and offer the following suggestions for its improvement. While the article addresses an important topic, major revisions are necessary to enhance its clarity, depth, and overall impact.

  1. Section Enhancement (Section 4): The article's Section 4 is notably brief and lacks the necessary depth to fully explore the connections between stress response, autophagy, and aging-related diseases. I strongly recommend expanding this section with more detailed explanations, relevant studies, and potential mechanisms that link these processes. This will provide readers with a comprehensive understanding of the complex interplay between stress response, autophagy, and their roles in aging-related diseases.

  2. Diagrams Improvement: Both diagrams included in the article need substantial improvement to better illustrate the discussed concepts. Consider providing more detailed visual representations that accurately convey the autophagy processes (Fig. 1) and proteostasis (Fig. 2). Clear labeling, appropriate captions, and a logical flow of information are essential to aid readers' comprehension.

  3. The article's organization and language need refinement for clarity. Ensure that each section flows logically from one to another, aiding readers in following the presented concepts. Additionally, use clear and concise language to explain complex concepts, making the information accessible to both experts and non-experts in the field.

  4. While the article touches upon the relationship between proteostasis pathways, longevity, and aging-related diseases, it lacks a more in-depth analysis of the specific studies, mechanisms, and evidence supporting these connections. Provide specific examples, experimental findings, and current research trends to bolster the claims made in the abstract. Additionally, a discussion about SIRT1 (Ref: https://pubmed.ncbi.nlm.nih.gov/32727328/) would also improve the content.

  5. Expand upon the article's concluding statements regarding the future implications of the discussed concepts. Delve into potential avenues for further research, the development of therapeutic interventions, and how these findings might contribute to advancements in addressing various diseases.

Minor corrections required

Author Response

Thank you for reviewing it carefully. I have made corrections to the manuscript according to the reviewer's feedback. I will provide the details below. Moreover, I have rechecked the English text and corrected any grammar errors.

1. Section Enhancement (Section 4): The article's Section 4 is notably brief and lacks the necessary depth to fully explore the connections between stress response, autophagy, and aging-related diseases. I strongly recommend expanding this section with more detailed explanations, relevant studies, and potential mechanisms that link these processes. This will provide readers with a comprehensive understanding of the complex interplay between stress response, autophagy, and their roles in aging-related diseases.

The section 4 has been expanded with additional related research. The added portions are indicated in red text.

2. Diagrams Improvement: Both diagrams included in the article need substantial improvement to better illustrate the discussed concepts. Consider providing more detailed visual representations that accurately convey the autophagy processes (Fig. 1) and proteostasis (Fig. 2). Clear labeling, appropriate captions, and a logical flow of information are essential to aid readers' comprehension.

I have made improvements to Figures 1 and 2 to enhance their readability.

3. The article's organization and language need refinement for clarity. Ensure that each section flows logically from one to another, aiding readers in following the presented concepts. Additionally, use clear and concise language to explain complex concepts, making the information accessible to both experts and non-experts in the field.

I have made revisions throughout the text regarding the points raised. The modified areas are indicated in red text.

4. While the article touches upon the relationship between proteostasis pathways, longevity, and aging-related diseases, it lacks a more in-depth analysis of the specific studies, mechanisms, and evidence supporting these connections. Provide specific examples, experimental findings, and current research trends to bolster the claims made in the abstract. Additionally, a discussion about SIRT1 (Ref: https://pubmed.ncbi.nlm.nih.gov/32727328/) would also improve the content.

I have revised the Section 5 based on the reviewer's advice. The modified areas are indicated in red text.

5. Expand upon the article's concluding statements regarding the future implications of the discussed concepts. Delve into potential avenues for further research, the development of therapeutic interventions, and how these findings might contribute to advancements in addressing various diseases.

I have enriched the "Conclusions" section to serve as a comprehensive summary of the topics discussed.

Reviewer 2 Report

Stress response and autophagy are interconnected cellular processes that play crucial roles in maintaining cellular homeostasis, promoting survival, and adapting to challenging conditions. When cells encounter stressors such as nutrient deprivation, oxidative stress, or protein misfolding, they activate stress response pathways to adapt and ensure survival. Autophagy is often upregulated as a part of this response. Furthermore, stress response mechanisms can induce autophagy as a means of quality control. For example, the unfolded protein response (UPR) triggered by endoplasmic reticulum (ER) stress can lead to the selective degradation of misfolded proteins through autophagy, a process known as ER-phagy or reticulophagy. Moderate stress exposure, such as caloric restriction or mild oxidative stress, can also induce autophagy and activate stress response pathways.

Overall, stress response and autophagy are interconnected processes that work in tandem to ensure cellular survival and maintenance in the face of various challenges. The activation of autophagy during stress helps cells adapt, clear damaged components, and promote longevity and health. Here in this review, authors successfully managed to describe the key aspects through the Role of Stress Response Pathway in Macroautophagy, Molecular Chaperones in CMA and Microautophagy, Proteostasis and Longevity, Stress Response and Autophagy Processes as Therapeutic Targets. The authors provided an overview of the critical genes and proteins involved in the process and also added valuable insight into therapeutics. Indeed, literature mining is impressive.

However, there are certain aspects that are missing from a reader's perspective. Here, I am offering some suggestions to improve the manuscript.

1) While reading the manuscript, it does not sound connected. That specific aspect is clearly missing in this review. Need to summarize each individual section independently. All sections are introduced but not concluded sufficiently to connect the next section. Authors must try to connect all the sections through important sentences and references, which will make this review more coherent and attractive.

2) Also, there are some minor corrections authors should consider:

a) The quality of the figures presented in the manuscript needs to be improved.

b) In Figure 1, the most appropriate would be to write “KFERQ-like motif” (PMID: 31150375; PMID: 33446043). Also, to be consistent in the figure, the components that undergo microautophagy could be marked in red.

c) Definitely, the authors must consider connecting Section 4 and Section 5 to make the manuscript more structured and linked. 

d) An additional table on Autophagy as a promoter of longevity, studied model organisms and references, could be an excellent addition to this review.

e) To provide a much broader aspect to the readers, authors could add some challenges in the field or challenging questions to be addressed, in a point-wise manner at the end.

Author Response

Thank you for providing valuable advice. I have revised the manuscript based on the reviewer's feedback. The corrected areas are highlighted in red text. Details are provided below.

1) While reading the manuscript, it does not sound connected. That specific aspect is clearly missing in this review. Need to summarize each individual section independently. All sections are introduced but not concluded sufficiently to connect the next section. Authors must try to connect all the sections through important sentences and references, which will make this review more coherent and attractive.

I have addressed the connection between the mentioned sections. Additionally, I have added conclusions. The revised portions are highlighted in red text.

2) Also, there are some minor corrections authors should consider:

a)The quality of the figures presented in the manuscript needs to be improved.

I have recreated the Figures to make them more readable.

b)In Figure 1, the most appropriate would be to write “KFERQ-like motif” (PMID: 31150375; PMID: 33446043). Also, to be consistent in the figure, the components that undergo microautophagy could be marked in red.

I have made the corrections as indicated.

c) Definitely, the authors must consider connecting Section 4 and Section 5 to make the manuscript more structured and linked. 

I have revised the connection between Section 4 and Section 5. The revised portions are indicated in red text.

d)An additional table on Autophagy as a promoter of longevity, studied model organisms and references, could be an excellent addition to this review.

I have added a new table for "Proteostasis and Longevity."

e) To provide a much broader aspect to the readers, authors could add some challenges in the field or challenging questions to be addressed, in a point-wise manner at the end.

I have also added content to the "Conclusions" section.

Round 2

Reviewer 1 Report

The authors have not followed the suggested changes which makes the manuscript not sufficient to be published. Please read the comments carefully. Especially comment no.s 1, 2 (reg. fig.1), and 5.

Author Response

I'm grateful for your review. I have included the points that the reviewer mentioned.

I have added the mechanisms for the activation of CMA and microautophagy due to stresses in Section 4. I have also included research on CMA and aging in Section 5. However, it was not possible to add research on microautophagy and longevity as there is very little available to date.  I have incorporated treatment-related perspectives in the Conclusions section that align with the theme of this manuscript. The added sections are indicated in red text.

I have accurately added labeling to Figures 1 and 2 to make them more reader-friendly. And, I have also included clear notations within the figures to make the processes more understandable.